# Spatial Prediction of High-Risk Areas for Asthma in Metropolitan Areas: A Machine Learning Approach Applied to Tehran, Iran

**Alireza Mohammadi** [1],*, **Elahe Pishgar** [2] **and Juan Aguilera** [3]

1   Department of Geography and Urban-Rural Planning, Faculty of Social Sciences, University of Mohaghegh Ardabili, Ardabil 5619911367, Iran
2   Department of Geography, Faculty of Geography, University of Tehran, Tehran 1417853933, Iran; epishagar2018@gmail.com
3   Center for Community Health Impact, UTHealth Houston School of Public Health, El Paso, TX 79902, USA
*   Correspondence: a.mohammadi@uma.ac.ir

**Abstract:** Asthma prevalence in large urban areas of developing countries is a significant public health concern, with increased rates driven by various socioeconomic and environmental factors. This study aims to predict asthma risk in Tehran, a major urban center in Iran. Data from 1473 asthma patients, alongside demographic, socioeconomic, air quality, environmental, weather, and healthcare access variables, were analyzed using geographic information systems (GIS) and remote sensing techniques. Three ensemble machine learning algorithms—Random Forest (RF), Gradient Boosting Machine (GBM), and Extreme Gradient Boosting (XGBoost)—were applied to model and predict asthma risk. A Negative Binomial Regression Model (NBRM) identified seven key predictors: population density, unemployment rate, particulate matter ($PM_{2.5}$ and $PM_{10}$), nitrogen dioxide ($NO_2$), sulfur dioxide ($SO_2$), neighborhood deprivation index, and road intersection density. Among the algorithms, GBM outperformed the others, with a training RMSE of 0.56 and a test RMSE of 1.07, demonstrating strong generalization. Additionally, GBM achieved the highest R-squared values (0.95 for training and 0.76 for testing) and lower MAE values (0.43 for training and 0.88 for testing). Effective pattern recognition was confirmed by EV values of 0.95 for training and 0.75 for testing, along with a Moran's I value of 0.17, indicating minimal spatial autocorrelation.

**Keywords:** large cities; spatial analysis; sociodemographic; built environment; ambient air pollution; urban heat islands; healthcare facilities; Sentinel-5; google earth engine; Landsat 8

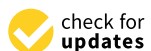

## 1. Introduction

Asthma, a primary non-communicable disease, affects both children and adults, with the highest prevalence among children. It causes inflammation and narrowing of the airways, leading to symptoms like coughing, wheezing, breathlessness, and chest tightness [1]. Approximately 300 million people have asthma all over the world, with 250,000 deaths attributed to the disease annually, most of which are preventable [2]. As of a 2023 study, asthma prevalence varied across continents: Asia (3.44%), Africa (3.67%), South America (4.90%), Europe (5.69%), North America (8.29%), and Oceania (8.33%); among global asthma cases, 26.70% were severe, 30.99% were eosinophilic, 48.95% included allergic rhinitis, and 7.0% to 25.40% included nasal polyps [3]. Asthma is often underdiagnosed and undertreated, particularly in low- and middle-income countries [1]. Various studies have

reported the prevalence of asthma in Iran in recent years at around 8.9% (95% confidence interval (CI): 8.5–9.3) [4,5].

Furthermore, demographic and socioeconomic factors like population density, aging, illiteracy, and unemployment are linked to higher rates of asthma prevalence and mortality in large urban areas [6]. High population density increases exposure to air pollution and asthma [7,8]. Aging populations may experience worsened asthma symptoms due to age-related changes [9,10]. Low literacy levels can hinder effective asthma management, leading to more severe outcomes [11]. Unemployment, often tied to lower socioeconomic status, limits access to healthcare and may increase exposure to asthma environmental triggers [12,13].

Previous studies have also shown that exposure to ambient air pollutants, including particulate matter ($PM_{2.5}$ and $PM_{10}$), nitrogen dioxide ($NO_2$), ozone ($O_3$), and sulfur dioxide ($SO_2$), is a significant risk factor for asthma prevalence [14–17] Particulate matter, consisting of tiny particles suspended in the air, can penetrate deep into the respiratory system, triggering airway inflammation and exacerbating asthma symptoms [18]. $NO_2$, primarily emitted from vehicles and industrial sources, has been associated with increased asthma incidence and severity, likely due to its irritant effects on the airways [19,20]. Increased levels of $O_3$, a highly reactive gas formed by the interaction of sunlight with pollutants like $NO_2$, were linked to asthma attacks and decreased lung function among individuals residing in urban centers [21]. $SO_2$, primarily emitted from industrial processes and power plants, can irritate the airways and exacerbate respiratory conditions such as asthma [20].

Large urban areas in developing countries also pose significant built environmental risks for asthma onset or fatality. A study has indicated that the prevalence, severity, and morbidity of asthma have significantly increased among residents in low-income urban areas [21]. Poor-quality urban environments, characterized by socioeconomic disadvantage, inadequate access to healthcare, and poor housing conditions, contribute to the exacerbation of asthma symptoms in deprived areas [22–25]. Road intersection density, often associated with high traffic volume and vehicular emissions, is another significant determinant of asthma prevalence [25,26]. The proximity of residential areas to busy roads increases exposure to traffic-related air pollutants, such as particulate matter and nitrogen oxides, which can aggravate asthma symptoms and respiratory inflammation [27].

According to previous studies [28,29], the Normalized Difference Vegetation Index (NDVI), a green space and vegetation density measure, has been inversely associated with asthma prevalence. Higher levels of greenery in neighborhoods are linked to improved air quality, reduced pollution exposure, and enhanced respiratory health outcomes [28,29]. Exposure to industrial emissions, including $SO_2$ and volatile organic compounds, is a known risk factor for asthma prevalence [30]. Additionally, proximity to fuel stations in urban areas, characterized by emissions from gasoline and diesel vehicles, is associated with increased asthma prevalence due to heightened exposure to benzene, a known respiratory irritant [31]. Urban heat islands (UHIs), where cities are hotter than surrounding areas due to human activities, worsen with climate change, leading to higher asthma prevalence and severity. Increased temperatures amplify air pollutants like ozone ($O_3$) and particulate matter ($PM_{2.5}$), which trigger asthma symptoms [32,33].

Climate change also prolongs heat waves and the pollen season, further aggravating asthma [34]. Urban areas experience high temperatures due to extreme climate conditions, worsening asthma by intensifying pollutants like ozone and particulate matter. Longer heat waves and extended pollen seasons also aggravate asthma [34]. Some studies confirmed that limited access to healthcare facilities in large cities of developing countries exacerbates asthma by delaying diagnosis, reducing treatment adherence, and hindering symptom management [35,36].

In Tehran, the capital city of Iran, it was estimated that the average asthma prevalence was 13.4% [37], exceeding the country's overall average asthma prevalence of 8.9%. Certain studies indicated that asthma among adults in Tehran was 11.73% [5]. In children, the highest reported prevalence was 32% in Tehran, while in adolescents, it was 37% [38]. Previous research on asthma in Tehran has frequently explored the correlation between asthma and patient lifestyle [39], the link between asthma and food allergies or environmental pollutants [40], and the economic implications [41] associated with asthma. Some studies have also examined asthma's spatial dimensions and spatial modeling within Tehran city using ensemble machine learning algorithms [42–44].

Spatial epidemiology is the application of theory and methods from epidemiology, geography, and statistics to describe spatial distributions of health outcomes and to analyze associations with possible causes to inform intervention and improve health [45]. Utilizing geographic information systems (GIS) for spatial analysis of asthma is essential for comprehending its prevalence in urban areas and informing prevention and management strategies [42,46,47]. Various geostatistical modeling techniques, including spatial regression [48], Bayesian [49], and machine learning algorithms (MLAs), integrate geographical and environmental data to identify patterns and correlations with asthma incidence [42].

Geographically Weighted Regression (GWR), Multiscale Geographically Weighted Regression (MGWR), Spatial Lag Model (SLM), Spatial Error Model (SEM), Bayesian Spatial Models, and Spatial Autoregressive Models (SAR) are primarily statistical or econometric models that incorporate spatial dependencies and structures [50]. These methods are valued for their interpretability and ability to model spatial relationships and dependencies explicitly [51]. However, they are not typically classified as machine learning algorithms (MLAs). Among the various machine learning algorithms, Random Forest (RF), Gradient Boosting Machine (GBM), and Extreme Gradient Boosting (XGBoost) are ensemble algorithms that have been specifically adapted for spatial count data. These algorithms capture complex, non-linear relationships, making them practical for regression and prediction tasks. While GBM improves accuracy by iteratively optimizing residual errors, RF minimizes overfitting by averaging multiple decision trees. Large spatial datasets benefit greatly from XGBoost's scalability and regularization. To balance bias, variance, and overfitting, grid search and cross-validation were used to optimize the hyperparameters for all algorithms, including the number of trees, learning rate, and regularization terms. These algorithms were selected because of their versatility and resilience in identifying significant predictors and interactions in spatial epidemiological settings [42,52].

Machine learning (ML), which offers improved disease prediction, risk assessment, and individualized treatment skills, has emerged as a key instrument in medical research in recent years. The ability of machine learning algorithms like Random Forest (RF), Gradient Boosting Machine (GBM), and XGBoost to capture intricate, non-linear correlations between risk factors is demonstrated by successful applications in domains including diabetes, cardiovascular illnesses, and cancer [53–55], as well as in the prediction of infectious diseases such as COVID-19 [56]. In the context of asthma, machine learning (ML) is an emerging method that enhances conventional statistical techniques by offering more precise forecasts and insights into the incidence of asthma. Although there are currently few studies on machine learning in asthma risk prediction [42–44], newer studies show how ML is increasingly used to advance knowledge and direct focused public health initiatives. By using cutting-edge machine learning techniques to forecast asthma risk in Tehran, this work adds to this trend while tackling the city's problems with extreme air pollution, urban heat island effects, and healthcare inequities. The study intends to further the integration of machine learning in asthma epidemiology by improving spatial risk prediction and informing public health initiatives using these methods.

This study fills essential gaps in the body of knowledge regarding the prevalence of asthma in Tehran. Although the prevalence of asthma in the city has been studied in previous years, machine-learning-based regression techniques for predicting spatial risk have not been widely applied. Furthermore, sociodemographic, built-environmental, and environmental factors—like air pollution, urban heat islands, and healthcare accessibility—have not all been thoroughly incorporated into a single spatial framework in previous studies. This study uses the Binomial Negative Regression Model (BNRM), geographic information systems (GIS), and sophisticated ensemble machine learning techniques, such as Random Forest (RF), Gradient Boosting Machine (GBM), and Extreme Gradient Boosting (XGBoost), to close these gaps. These resources pinpoint high-risk areas and offer helpful information for focused public health initiatives. The study uses extensive spatial regression models and a larger dataset to improve prediction accuracy and policy relevance. Tehran is a critical case study because of its extreme air pollution, notable urban heat island effects, and inequalities in access to healthcare. Asthma risks are increased by the city's congested urban environment and industrial pollution, underscoring the necessity of efficient spatial modeling. This study contributes to a more thorough understanding of asthma prevalence by concentrating on Tehran and provides a solid framework for predicting spatial risk in highly polluted urban environments. It focuses on combining socioenvironmental factors to identify high-risk locations, which will help direct focused public health programs in the city.

## 2. Materials and Methods

### 2.1. Study Setting

Tehran, Iran's capital, covers 730 square kilometers with a population density of about 11,000 per square kilometer. It has over 8 million residents within the city and more than 15 million in the metropolitan area [57]. The city, divided into 22 districts and 350 neighborhoods, provides a unique environment for studying asthma prevalence and risk factors. The city experiences a semi-arid climate marked by hot summers and cold winters, resulting in diverse seasonal air quality challenges [58,59]. Tehran's urban heat island effects are significant, with temperature variations of up to 7 °C between urban and rural areas, influenced by dense construction and limited green spaces [60]. Approximately 7.5% of Tehran's population is 65 years or older, the illiteracy rate is 7%, and the unemployment rate was reported to be about 7% [61].

Tehran is one of the world's most polluted cities. It faces significant air pollution challenges, with elevated levels of particulate matter ($PM_{2.5}$ and $PM_{10}$) exceeding safe thresholds, reaching 30–50 $\mu g/m^3$ for $PM_{2.5}$ and 70–100 $\mu g/m^3$ for $PM_{10}$ during peak pollution seasons [62]. Additionally, $NO_2$ levels persistently average 50–60 $\mu g/m^3$, surpassing the WHO limit of 40 $\mu g/m^3$. Ozone ($O_3$) concentrations can spike to 150 $\mu g/m^3$ in summer due to photochemical reactions from vehicular and industrial emissions, while $SO_2$ levels are generally lower, averaging 20–30 $\mu g/m^3$ with occasional industrial spikes [63,64].

According to calculations based on OSM data using a geographic information system, Tehran's urban infrastructure showcases a high road intersection density, notably exceeding 400 intersections per square kilometer in central districts. Furthermore, according to the calculation based on Landsat 8 images, the city's average Normalized Difference Vegetation Index (NDVI) was 0.3, indicating moderate vegetation cover, with higher values observed in the greener northern areas. The city has 6.5 $m^2$ of green area for every person [65]. Proximity to industrial zones, particularly in the outskirts, exposes residents to significant industrial emissions [66]. There are about 120 public and private hospitals in Tehran, and about 60 hospitals provide services to asthma patients [61]. However, the spatial distribution of these hospitals is uneven, and local access to them varies. Figure 1 illustrates

the location of the study area, the distribution of hospitals that have admitted patients with asthma, and the spatial distribution of asthma cases from 2020 to 2023.

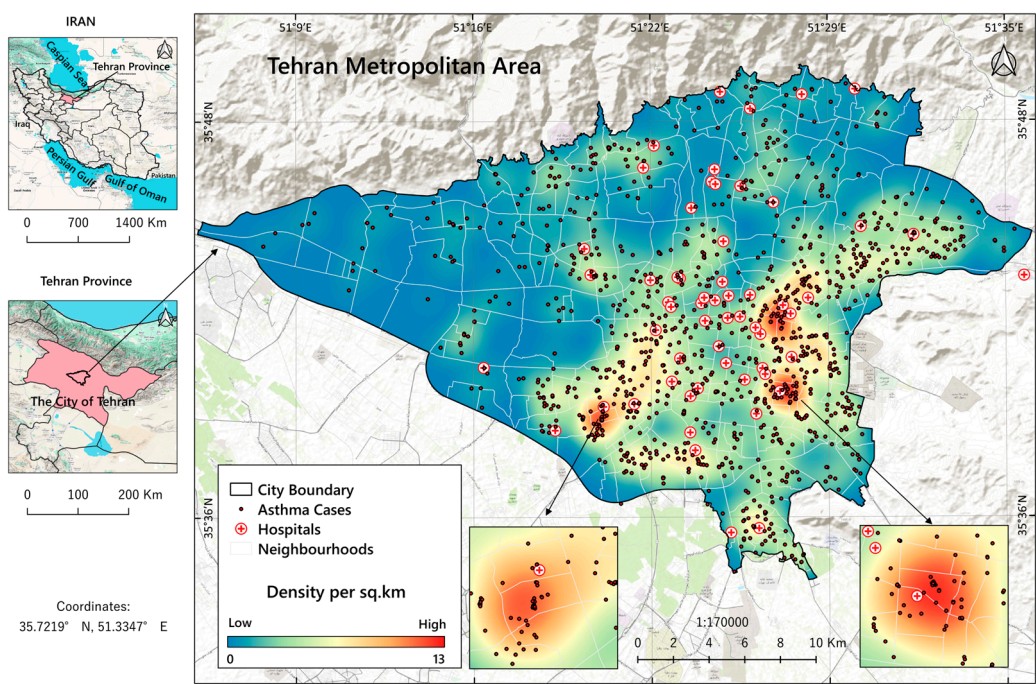

**Figure 1.** Spatial distribution map of asthma cases and their density in the study area.

## 2.2. Data Source and Its Processing

We employed a comprehensive dataset containing spatial and non-spatial data, consisting of 1 dependent variable and 15 independent variables selected based on existing literature. This dataset can be categorized into four primary groups: (1) census datasets, (2) remote sensing data acquired from Landsat 8 and Sentinel-5 satellite products, (3) GIS datasets, and (4) open-source datasets, which incorporate spatial data obtained from OpenStreetMap. The asthma data (response variable) consisted of information on 2179 patients, collected in Excel format from 70 hospitals affiliated with the Ministry of Health and Medical Education (MHME), spanning the period from 6 July 2020, to 2 August 2023, with an assigned ethics code number.

After incomplete or outlier cases were eliminated, 1473 patients made up the final sample for analysis. To protect patient privacy, the data were geocoded using the UTM coordinate system, which represents locations as neighborhood-level point features. This method allowed for spatial analysis while maintaining patient confidentiality. This secondary data analysis was approved as ethical and per the Ministry of Health and Medical Education's guidelines. The confirmed asthma diagnoses in the area during the designated study period are represented by the asthma cases included in this analysis. The study's reproducibility was ensured by adhering to a protocol that described the data integration and spatial analysis methodology. Although the data offer insightful information about the distribution of asthma, this study does not purport to provide a thorough epidemiological evaluation of all asthma cases in the study area.

Figure 2 shows the research methodology flowchart.

Patient information included age, sex, date of admission, date of discharge, hospitalization outcome (discharged, complete recovery, partial recovery, and death), and patient address. We used the patient addresses from the original dataset to create the point data for the dependent variable. These addresses were geocoded to determine each case's exact geographic coordinates (latitude and longitude). Following this, the data were aggregated

into 350 neighborhoods within Tehran city and compiled as asthma cases (N = 1473) (dependent variable) in a shapefile. The indicators considered are provided in Table 1, based on existing literature and data availability for the city. Population data and data on elderly individuals (aged 65 and older) were extracted from the 2016 census data [67]. The proportion of illiterate people (%) among individuals aged 6 and above and the proportion of unemployed people (%) among individuals in the labor force aged 15 to 64 were calculated for the 350 neighborhoods using census data. Additionally, information on the neighborhood deprivation index (percentage of deteriorated buildings in each neighborhood) was obtained from Tehran Municipality, indicating areas with deprived buildings.

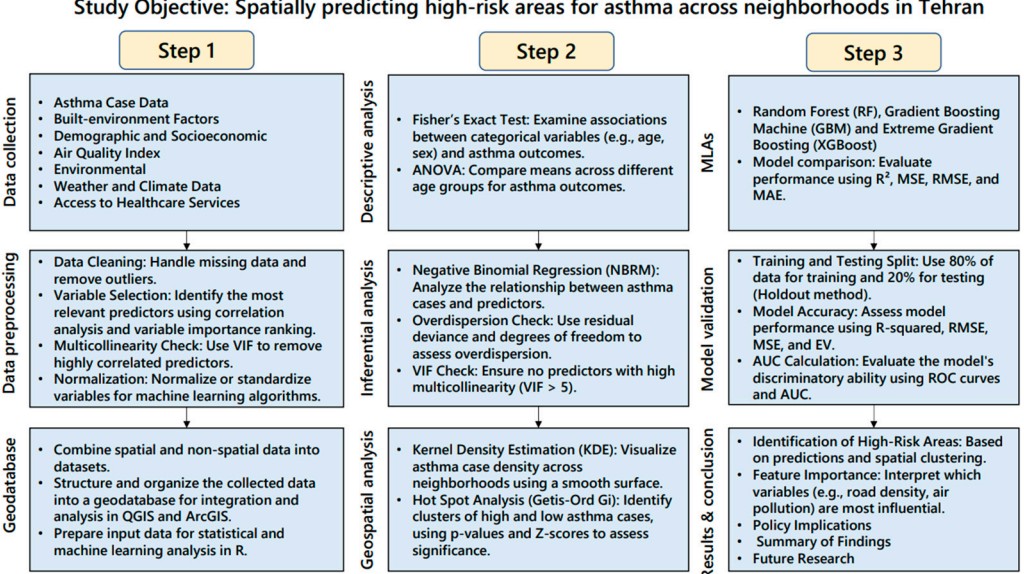

**Figure 2.** Research methodology flowchart.

**Table 1.** Indicators identified from the literature review to examine the relationship between asthma prevalence and neighborhood characteristics in Tehran (2020–2024).

| Aspects | Indicator | Spatial Database and Data Type | Source |
|---|---|---|---|
| Demographic and Socioeconomic | V1: Population density (per sq.km) | Census, ESRI shapefile | [67] |
| | V2: Proportion of elderly (%) | Census, ESRI shapefile | [67] |
| | V3: Proportion of illiterate people (%) | Census, ESRI shapefile | [67] |
| | V4: Proportion of unemployed people (%) | Census, ESRI shapefile | [67] |
| Air Quality Index | V5: Particulate matter (AAI include $PM_{2.5}$ and $PM_{10}$) | Sentinel-5, Raster | Google Earth Engine |
| | V6: Nitrogen dioxide ($NO_2$) | Sentinel-5, Raster | Google Earth Engine |
| | V7: Ozone ($O_3$) | Sentinel-5, Raster | Google Earth Engine |
| | V8: Sulfur dioxide ($SO_2$) | Sentinel-5, Raster | Google Earth Engine |
| Environmental | V9: Neighborhood deprivation index (%) | Land use map, ESRI shapefile | Tehran municipality, OpenStreetMap |
| | V10: Road intersection density (per square kilometers) | OSM, ESRI shapefile, and Raster | OpenStreetMap |
| | V11: Normalized Difference Vegetation Index (NDVI) | Landsat 8, Raster | Google Earth Engine |
| | V12: Exposure to industrial emissions | Land use map, OSM, ESRI shapefile | OpenStreetMap |
| | V13: Proximity to fuel stations | Land use map, OSM, ESRI shapefile | Tehran Municipality, OpenStreetMap |
| Weather and Climate | V14: Urban heat islands (UHIs) | Landsat 8, Raster | Google Earth Engine |
| Access and Utilization of Healthcare Services | V15: Access to healthcare facilities | Land use map, OSM, ESRI shapefile | Tehran municipality, OpenStreetMap |

Data on the most common air pollutants, including offline high-resolution imagery of the UV Aerosol Index (UVAI), also known as the Absorbing Aerosol Index (AAI), and levels of nitrogen dioxide ($NO_2$), ozone ($O_3$), and sulfur dioxide ($SO_2$), were extracted using the Sentinel-5 Precursor, a satellite launched on 13 October 2017 by the European Space Agency to monitor air pollution. We employed the Google Earth Engine (GEE) cloud-based geospatial analysis platform to extract all city-level pollutant concentrations with a uniform cell size of 10 m. To calculate road intersection density (per square kilometer), we extracted all intersections from the road network using OpenStreetMap (OSM) data with QGIS software (version 3.36.3). The Normalized Difference Vegetation Index (NDVI) is a quantitative index of greenness ranging from 0 to 1, where 0 represents minimal or no greenness, and 1 represents maximum greenness [68]. The Normalized Difference Vegetation Index (NDVI) formula is typically represented as:

$$\text{NDVI} = \frac{(\text{NIR} - \text{Red})}{(\text{NIR} + \text{Red})} \tag{1}$$

NIR denotes near-infrared reflectance, while Red signifies red reflectance. NDVI is a pivotal metric within remote sensing analytical products employed for vegetation assessment [69]. NDVI was determined using Landsat 8 imagery. For Landsat 8, the NDVI calculation is expressed as (Band 5 − Band 4)/(Band 5 + Band 4) [70].

Data regarding the locations of industrial sites (V12), fuel stations (V13), and healthcare facilities (V15) were extracted from the city's land use vector map, obtained from the Tehran City Municipality. To calculate V12, we computed the spatial density of industrial units per square kilometer. For variable V13, we used the Euclidean distance method to calculate the distance (in meters) between the neighborhood center and fuel stations. The proximity of these locations to the 350 neighborhoods' centroids was assessed using Euclidean distance tools in ArcGIS Pro. To calculate V15, we computed the spatial density of health centers (such as hospitals) per square kilometer. These data were subsequently cross-referenced with OpenStreetMap datasets.

Urban heat islands (UHIs) are characterized by elevated temperatures in urban areas compared to their rural surroundings, primarily due to human activities. Land surface temperature (LST) is a key metric for identifying UHIs, with urban areas typically exhibiting higher LSTs due to heat-absorbing surfaces and reduced vegetation cover [71]. The UHIs were quantified using the USGS Landsat 8 Level 2, Collection 2, Tier 1 product via Google Earth Engine (GEE). A commonly used formula for LST calculation from satellite imagery is derived from Planck's law, which relates the radiance detected by the sensor to the surface temperature. The formula is expressed as follows [68,72]:

$$\text{LST} = \frac{K_2}{\ln\left(\frac{K_1}{T_B} + 1\right)} - 273.15 \tag{2}$$

where LST represents the land surface temperature in degrees Celsius, and $K_1$ and $K_2$ denote calibration constants specific to the sensor utilized. $T_B$ signifies the brightness temperature recorded by the sensor. This formula stems from the fundamental principles of Planck's law and the Stefan–Boltzmann law, elucidating the correlation between the emitted radiance from a surface and its temperature [68,72].

Particulate matter (AAI, including $PM_{2.5}$ and $PM_{10}$) (V5), nitrogen dioxide ($NO_2$) (V6), ozone ($O_3$) (V7), sulfur dioxide ($SO_2$) (V8), and the Normalized Difference Vegetation Index (NDVI) (V11) and urban heat islands (UHIs) (V14) from Landsat 8 were among the environmental and air quality variables that we extracted using Google Earth Engine. Mean values were compiled for every spatial unit after raster data had been pre-processed

to match neighborhood polygons spatially. We used the remote sensing variables' annual average values for compatibility with the 2020–2023 aggregated asthma data. Additionally, using the Sentinel-5 data's temporal coverage and spatial consistency, the spatial–temporal relationships between these variables and asthma prevalence were examined.

After preparing the spatial layers for each predictive variable, all selected spatial data indicators were transformed to the UTM Projection, WGS-84 Datum, Zone 39 N, with a uniform cell size of 30 m resolution to prevent any spatial output errors. The values of each variable (averaging where necessary) were then extracted for each neighborhood and stored in a geodatabase using ArcGIS Pro, maintaining the same coordinate system.

Figure 3 illustrates the spatial distribution of predictor value ranges across the study area, collected at the neighborhood level, which serves as the spatial analysis unit.

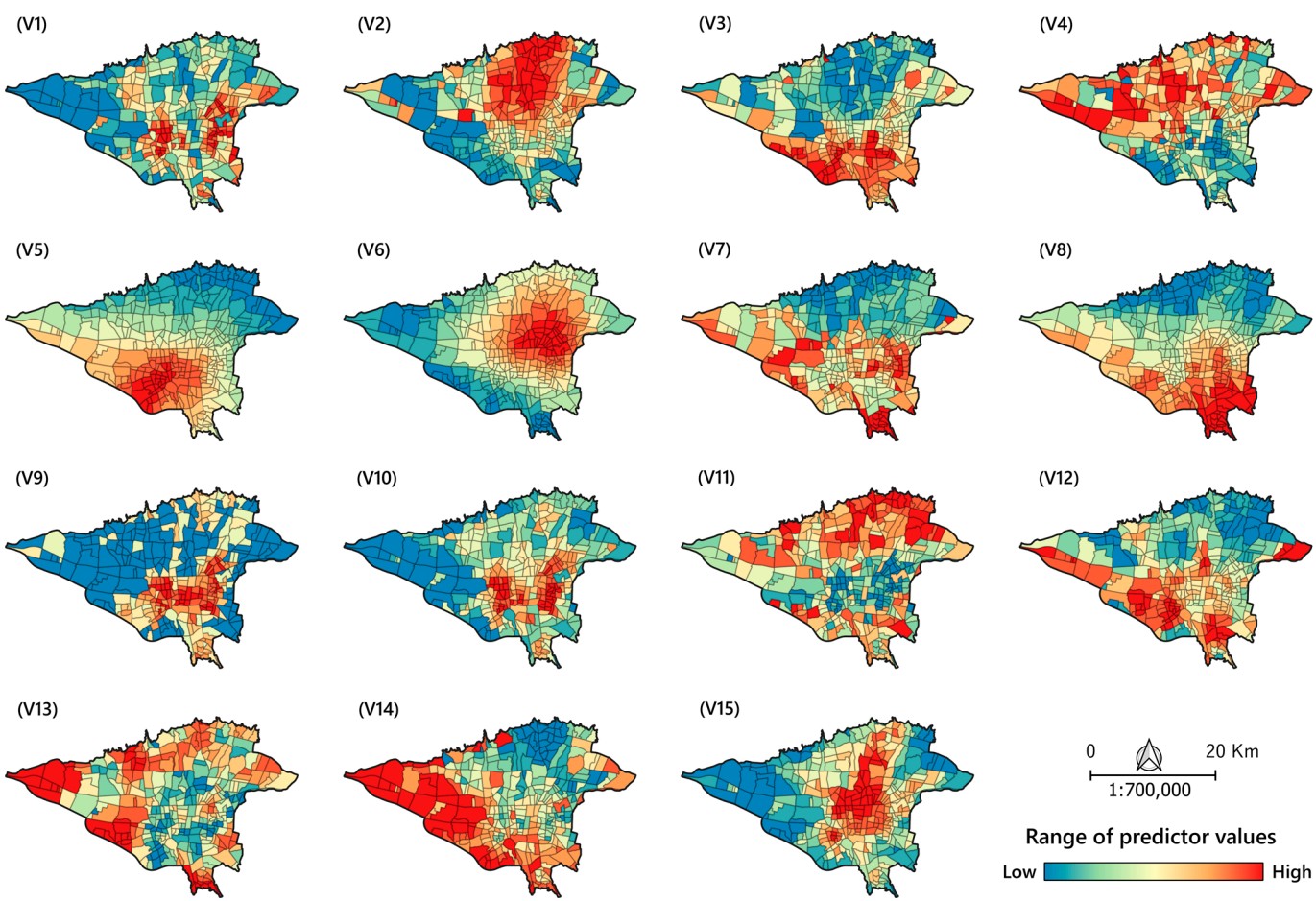

**Figure 3.** Spatial distribution maps of initial predictors in the study area.

### 2.3. Analytical Methods

### 2.3.1. Statistical Methods Used for Descriptive Analysis

We employed Fisher's exact test, an extension of the Chi-squared analysis [73], to evaluate the association between age, sex, and disease outcome to determine their independence under the null hypothesis. Additionally, analysis of variance (ANOVA) serves as a valuable statistical method for assessing disparities among the means of three or more groups, akin to an extension of the t-test for comparing multiple independent samples [74]. In this study, ANOVA was employed to assess the differences in mean ages across distinct age groups—categorized as "children" (age < 12), "adolescents" (age between 12 and 17), "adults" (age between 18 and 59), and "elderly" (age 60 and above)—concerning various disease outcome categories.

### 2.3.2. Statistical Methods Used for Inferential Analysis
Negative Binomial Regression Model (NBRM)

There are various methods to detect localized influential predictors in regression analysis, such as Explanatory Regression (ER), Ordinary Least Squares (OLS), and Generalized Linear Models (GLMs) [75]. GLMs are particularly noted for their effectiveness in pinpointing the most significant predictors in count data, especially when overdispersion is present. We assessed overdispersion by comparing the residual deviance to the degrees of freedom.

While machine learning algorithms are better than simple regressors in fitting data with multicollinearity, multicollinearity can affect model interpretability and feature importance across different machine learning algorithms [76]. Therefore, first, we ran a multicollinearity test to find predictors with high variance inflation factor (VIF) issues in R-Studio using the "car" package [77]. We removed highly correlated predictors according to previous studies (VIF > 5) [42]. We removed only one variable (V14) with the highest correlation (VIF = 5.2).

In the next step, Negative Binomial Regression (NBRM) was employed using the "MASS" package in R-Studio [78] to examine the relationship between asthma cases and predictors in this study. This method is ideal for counting data with overdispersion, a common issue in ecological and health studies [79]. NBRM extends Poisson regression to accommodate this variability, providing a better fit and more reliable estimates of coefficients and inference. In the generalized linear model framework, NBRM (Equation (3)) models the total count of events (Y) within a defined space–time interval, parameterized from a Poisson–gamma mixture as described by Hilbe [80], or equivalently, as the count of failures before achieving the $(1/\alpha)$th success. The NBRM model can be written as [81]:

$$\Pr(y_i|x_i) = \frac{\Gamma\left(y_i + \alpha^{-1}\right)}{y_i!\Gamma(\alpha^{-1})} \left(\frac{\alpha^{-1}}{\alpha^{-1} + \mu_i}\right)^{\alpha^{-1}} \left(\frac{\mu_i}{\alpha^{-1} + \mu_i}\right)^{y_i} \tag{3}$$

The formula represents the probability mass function of Negative Binomial Regression (NBRM), where $\Pr(y_i|x_i)$ denotes the probability of observing $y_i$ events given predictor variables $x_i$. The function incorporates parameters $\alpha$ and $\mu_i$, which describe the dispersion and mean, respectively. This equation is crucial for modeling count data with overdispersion, accommodating scenarios where the variance exceeds the mean, as commonly encountered in ecological and health studies [81].

### 2.3.3. Geospatial and Spatial Statistics Methods for Spatial Analysis
Kernel Density Estimation (KDE)

A well-known quartic type of kernel density estimation (KDE) method [82] was applied to create a heat map with a 30-square-meters cell size resolution, mapping asthma cases within the study area. This method assesses asthma case density per square kilometer using a smooth, continuous surface fitted over observed data points. It employs a quartic (biweight) kernel to enhance spatial visualization and capture patterns effectively, enabling detailed mapping and nuanced modeling of event distribution [83]. The optimal bandwidth size was determined using the mean random distance (RD mean) method [84]. A bandwidth of 1000 m, based on RD mean calculations, was selected for its effectiveness in producing a smooth density map.

The Hot Spot Analysis (Getis-Ord Gi*)

The Getis-Ord Gi* statistic, known as hot spot analysis (HAS), is a cluster mapping technique employed in health-related analysis to examine event density in specific locations [85]. Hot spots are areas with concentrated incidents vital for disease prevention [86]. The Gi* statistic identifies local spatial autocorrelation through Z score and *p*-value, in-

dicating clusters of high or low values [86]. Significant Z scores arise when a feature's local sum and neighbors differ significantly from the overall sum, suggesting non-random clustering [87]. Using point feature patterns, the Gi* statistic has been widely used to pinpoint hot and cold spots [88]. This study utilized the Getis-Ord Gi* statistic to detect hot and cold spots of asthma occurrences based on sample data at the neighborhood level, applying the "K nearest neighbors" (KNN) method for spatial relationship conceptualization [86]. In spatial analysis, K nearest neighbors (KNN) identify the K closest polygons to a target based on metrics like Euclidean distance, which are ideal for detecting clusters or patterns involving neighboring polygons. This method is flexible, allowing us to adjust K for tailored proximity analysis and understand spatial relationships among features [85,86].

Our analysis used the false discovery rate (FDR), a statistical technique to manage false positives among significant outcomes during multiple-hypothesis testing. By lowering critical *p*-values, FDR addresses the heightened likelihood of false positives due to multiple comparisons, thereby enhancing accuracy in identifying genuinely significant findings in extensive datasets or spatial analyses [85,86].

The KDE creates a smooth surface to visualize event density, highlighting areas of high concentration without assessing statistical significance. At the same time, HAS (Getis-Ord Gi*) uses Z-scores and *p*-values to identify statistically significant clusters of high and low values, providing an analytical approach to detect local spatial autocorrelation [86].

### 2.3.4. Methods for Spatial Predictions Using MLAs

Among various types of machine learning algorithms, Random Forest (RF), Gradient Boosting Machine (GBM), and XGBoost algorithms were chosen for regression and prediction analysis. These MLAs are well-suited for spatial regression analysis due to their ability to manage complex relationships, large datasets, and noise. They enhance predictive accuracy, reveal key factors, and offer robust, generalizable insights into asthma prevalence [43,44,89].

### Random Forest (RF)

Random Forest Regression (RF) is an ensemble-supervised machine learning algorithm developed by Leo Breiman and Adele Cutler, which creates models and generates predictions using an adaptation of the Random Forest algorithm [90]. The RF algorithm was designed and formulated for regression and prediction tasks. It constructs multiple decision trees using the bagging technique with bootstrapped samples, which involves generating random samples from the input data and training decision trees on these samples [91]. Developed by Ho in 1995 [92] and extended by Breiman in 2001 [93], RF is valued for its ease of implementation, speed, and high performance. For regression, it predicts by averaging the outputs of individual trees, reducing overfitting through majority voting. The algorithm ensures uncorrelated decision trees by selecting random feature subsets for training, which reduces model variance and enhances prediction accuracy, making it a robust choice for regression and prediction tasks. If implemented accordingly, RF can be adapted for count data, primarily through Poisson regression trees [91]. The RF algorithm constructs a model using a bagging technique, where multiple decision trees are created in parallel with random subsets of the training data. Each tree votes on an outcome, and the RF algorithm aggregates these votes for prediction. This ensemble method addresses overfitting issues of individual trees, resulting in a robust and intuitive model that requires fewer parameters [90].

### Gradient Boosting Machine (GBM)

GBM is a robust machine learning algorithm used for regression tasks. It builds an ensemble of decision trees sequentially, where each new tree corrects the errors made by the

previous ones by fitting them to the residuals. This technique uses weak learners (simple decision trees) and employs the mean squared error loss to improve the model progressively. Key features of GBM include the learning rate, which controls the contribution of each tree, and regularization techniques that prevent overfitting by limiting tree depth and applying penalties [94]. The algorithm also incorporates early stopping, monitoring performance on a validation set to halt training when improvement ceases while providing extensive hyperparameter tuning options for precise control over model complexity. Known for its high accuracy and ability to handle large datasets [95], GBM is highly effective in various practical applications. It offers superior performance in data-driven tasks and competitive machine learning challenges [96].

Extreme Gradient Boosting (XGBoost)

The XGBoost is an advanced implementation of GBM, recognized for its efficiency, flexibility, and optimization. It excels in handling sparse data, leveraging parallel processing, and utilizing weighted quantile sketching to enhance accuracy and scalability on large datasets. XGBoost supports custom loss functions, integrates built-in cross-validation, and is extensively used in competitive environments and industry applications for classification, regression, and ranking, owing to its strong performance and rich feature set [97].

MLA Implementation Procedure

Packages "randomForest" [93], "gbm", and "xgboost" [97] were used in the R-Studio software (Version 2024.12.0+467), respectively, to run the algorithms. Similar settings were applied when running the algorithms to perform the same analyses and compare the model outputs. We uploaded the input database, including the six most significant predictors and our response variable (asthma cases), to the algorithms in R-Studio as a comma-separated file (*.csv). Next, we split the data into training and test sets to evaluate the model's performance. We set 100 as the number of trees for each algorithm. To validate MLAs, 80% of the data is typically allocated for training, while the remaining 20% is reserved for testing using the holdout method. The holdout method is a technique in machine learning in which the dataset is split into separate training and testing sets to evaluate model performance [98].

MLAs Accuracy Assessment

The most commonly used validation metrics for count data in MLAs, including R-squared ($R^2$), root mean squared error (RMSE), mean squared error (MSE), mean absolute error (MAE), and explained variance (EV) or variance explained (VE), were applied to measure model performance and select the best model for analysis and interpretation [99]. R-squared ($R^2$) in machine learning measures how well a regression model fits the data, with values ranging from 0 to 1 indicating poor to perfect fit, respectively [99]:

$$R^2 = 1 - \frac{\sum_{i=1}^{n}(predicted_i - actual_i)^2}{\sum_{i=1}^{n}(actual_i - mean(actual))^2} \quad (4)$$

MSE is a metric used in regression analysis to measure the average squared differences between predicted and actual values. It provides a way to quantify the overall quality of a model's predictions, where lower MSE values indicate better performance [99]:

$$MSE = \frac{1}{n}\sum_{i=1}^{n}(predicted_i - actual_i)^2 \quad (5)$$

where the number of observations is represented by *n*.

RMSE in regression quantifies average prediction errors by taking the square root of the average squared differences between predicted and actual values, indicating prediction accuracy. Lower RMSE values signify better alignment between predicted and actual outcomes, indicating superior model performance in minimizing prediction errors [100]:

$$RMSE = \sqrt{\frac{1}{n}\sum_{i=1}^{n}(predicted_i - actual_i)^2} \qquad (6)$$

MAE is a metric that measures the average absolute differences between predicted and actual values. It provides a straightforward way to quantify the magnitude of errors in a model's predictions, where lower MAE values indicate better predictive accuracy [99]:

$$MAE = \frac{1}{n}\sum_{i=1}^{n}|predicted_i - actual_i| \qquad (7)$$

In statistics, explained variation (EV) measures the proportion to which a mathematical model accounts for a given dataset's variation (dispersion). EV measures how well predictors (in regression) account for variability in the response variable or total dataset. Higher EV values indicate greater explanatory power, which is crucial for assessing model effectiveness or dimensionality reduction success [101]. We applied all suitable metrics to measure MLA's performance using the "caret" package designed for MLAs [102].

After applying the MLAs, the Global Moran's Index (GMI) was employed to assess spatial autocorrelation in model residuals, offering insights into potential spatial dependencies. This index is instrumental in identifying whether the residuals demonstrate significant spatial clustering or dispersion across the study area, which aids in selecting the most suitable model. The GMI is defined as follows [86,90]:

$$I = \frac{n\left(\sum_{i=1}^{n}\sum_{j=1}^{n} w_{ij}(x_i - \overline{x})(x_j - \overline{x})\right)}{\left(\sum_{i=1}^{n}\sum_{j=1}^{n} w_{ij}\right)\left(\sum_{i=1}^{n}(x_i - \overline{x})^2\right)} \qquad (8)$$

where *n* represents the total number of spatial units (in this case, the number of neighborhoods in the Tehran metropolitan area); $x_i$ denotes the standardized death rate of overall cancers per 100,000 people in the neighborhood; *I* is the mean death rate across all counties; and $w_{ij}$ represents the spatial weight between neighborhood *i* and *j*. Moran's Index (I) ranges from −1 to +1, with values further from zero indicating stronger (positive or negative) spatial autocorrelation [103]. To compute the GMI, training and test data residuals were assigned as value fields for each study unit. This was followed by applying the GMI using GeoDa software (Version 1.22.0.4) [104].

## 3. Results

### 3.1. Non-Spatial Descriptive Findings

According to the sample data's temporal trend of asthma prevalence, 1473 cases occurred over four years. Just two cases, or 0.14% of the total, were reported in 2020. This rose sharply to 823 cases (55.86%) in 2022 after increasing significantly to 46 cases (3.12%) in 2021. In 2023, there was a decrease, though, with 602 cases (40.88%). Data analysis revealed that the average age of the patients was 53 years. Among all cases, 47.5% (n = 699) were female, and 52.5% (n = 774) were male. Among all cases (N = 1473), 2.02% (n = 28) were adolescents, 38.6% (n = 569) were adults, 12.61% (n = 186) were children, and 46.77% (n = 690) were elderly. Examining the relationship between age groups and asthma using Fisher's exact test showed a significant (*p*-value < 0.00) relationship between age

groups and asthma. However, the results indicated that the relationship was not significant (*p*-value > 0.05) between gender and asthma prevalence, showing no substantial difference in the disease prevalence between genders. The average length of hospital stay of the patients was five days. The time distribution of the data shows that nearly 60% of the patients were admitted from February to July. Upon analysis, it was found that within the studied samples, 41.61% (n = 614) had achieved complete recovery, 11.12% (n = 163) had passed away, 11.83% (n = 172) were discharged, and 35.54% (n = 524) had partially recovered. The results of the ANOVA test indicated that the age group variable has a highly significant effect on the outcome of death ($p < 2 \times 10^{-16}$). Among the total deaths due to asthma (614 cases), 53% (n = 85) were men. Notably, the highest proportion of mortality was observed among the elderly age group (8.81%), followed by adults (2.24%).

### 3.2. Spatial Analysis Findings

#### 3.2.1. KDE Method Results

In a study of 350 neighborhoods, the KDE map revealed significant variability in asthma cases. The mean KDE value is 3.4, with a standard deviation of 2.38 and a range of 11.73. The maximum and minimum KDE values are 11.77 and 0.04, respectively. Notably, 161 neighborhoods (approximately 46%) have KDE values exceeding 3.4 cases per square kilometer, indicating higher asthma burdens in these areas (see Figure 1).

#### 3.2.2. Hot Spot Analysis Results

The "hot spot" analysis identified 87 hot spots (24.9%, with *p*-value < 0.05 and Gi* statistic > 1) and 88 cold spots (25.1%, with *p*-value < 0.05 and Gi* statistic > −1). Hot spots have significantly higher asthma cases, while cold spots have significantly lower concentrations. The spatial distribution of these spots reveals that areas with hot spots and high asthma burden are in the west and east of the city center, indicating localized clusters of high asthma incidence (see Figure 4).

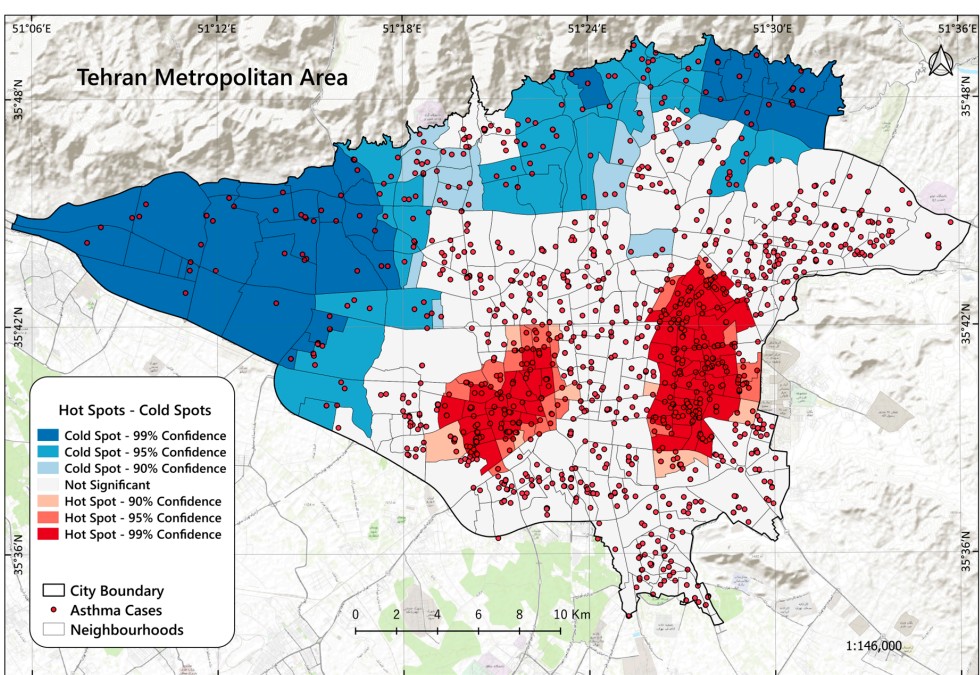

**Figure 4.** The spatial distribution of asthma case hot spots and cold spots within the study area.

### 3.3. Results from Negative Binomial Regression Model (NBRM)

In our analysis of the factors affecting the distribution of asthma cases in Tehran, we employed a Negative Binomial Regression Model (NBRM) to account for overdispersion

in the count data. The predictors included variables V1 through V15, with the response variable being the count of asthma cases. We calculated each predictor's variance inflation factor (VIF) to address multicollinearity. A VIF value exceeding 5 indicates potential multicollinearity issues. Predictors with VIF values greater than 5, such as V14, were removed iteratively. The final set of predictors included in our NBRM were free of significant multicollinearity. We assessed overdispersion by comparing the residual deviance to the degrees of freedom. The ratio of deviance to degrees of freedom was 0.632, indicating no significant overdispersion in the model. We employed a stepwise selection method to refine the model further. The final model, selected based on the lowest Akaike information criterion (AIC), is summarized in Table 2.

**Table 2.** Initial summary of negative binomial regression model coefficients.

| Predictor | Estimate | Std. Error | z Value | Pr (>\|z\|) |
|-----------|----------|------------|---------|-------------|
| V1  | $1.9 \times 10^{-5}$  | $4.9 \times 10^{-6}$ | $3.8 \times 10^{0}$   | $1.5 \times 10^{-4}$ *** |
| V2  | $-3.2 \times 10^{-2}$ | $1.9 \times 10^{-2}$ | $-1.6 \times 10^{0}$  | $1.0 \times 10^{-1}$ |
| V3  | $8.8 \times 10^{-3}$  | $1.1 \times 10^{-2}$ | $7.7 \times 10^{-1}$  | $4.4 \times 10^{-1}$ |
| V4  | $-3.2 \times 10^{-2}$ | $4.1 \times 10^{-2}$ | $-7.7 \times 10^{-1}$ | $4.4 \times 10^{-1}$ |
| V5  | $1.4 \times 10^{0}$   | $6.2 \times 10^{-1}$ | $2.2 \times 10^{0}$   | $2.6 \times 10^{-2}$ * |
| V6  | $2.3 \times 10^{3}$   | $5.9 \times 10^{2}$  | $3.9 \times 10^{0}$   | $1.1 \times 10^{-4}$ *** |
| V7  | $-1.4 \times 10^{2}$  | $2.2 \times 10^{2}$  | $-6.3 \times 10^{-1}$ | $5.3 \times 10^{-1}$ |
| V8  | $5.2 \times 10^{3}$   | $1.6 \times 10^{3}$  | $3.3 \times 10^{0}$   | $1.1 \times 10^{-3}$ ** |
| V9  | $-5.6 \times 10^{-3}$ | $2.2 \times 10^{-3}$ | $-2.6 \times 10^{0}$  | $1.0 \times 10^{-2}$ * |
| V10 | $5.9 \times 10^{-4}$  | $1.5 \times 10^{-4}$ | $4.0 \times 10^{0}$   | $7.6 \times 10^{-5}$ *** |
| V11 | $9.5 \times 10^{-1}$  | $1.1 \times 10^{0}$  | $8.7 \times 10^{-1}$  | $3.8 \times 10^{-1}$ |
| V12 | $3.4 \times 10^{-3}$  | $4.6 \times 10^{-3}$ | $7.4 \times 10^{-1}$  | $4.6 \times 10^{-1}$ |
| V13 | $-2.1 \times 10^{-5}$ | $3.6 \times 10^{-5}$ | $-5.7 \times 10^{-1}$ | $5.7 \times 10^{-1}$ |
| V14 | $-2.7 \times 10^{-2}$ | $4.1 \times 10^{-2}$ | $-6.6 \times 10^{-1}$ | $5.1 \times 10^{-1}$ |
| V15 | $6.7 \times 10^{-3}$  | $2.3 \times 10^{-2}$ | $2.9 \times 10^{-1}$  | $7.7 \times 10^{-1}$ |

Significance codes: 0—'***', 0.001—'**', 0.01—'*', 0.05—'.', 0.1—' ', 1, AIC = 1219.

The lack of significant overdispersion and multicollinearity issues validated the model's robustness. The stepwise selection process further optimized the model, ensuring the inclusion of only the six most relevant predictors. The final model emphasizes several key factors that significantly impact the distribution of asthma cases in Tehran. Variables V1 (population density), V4 (proportion of unemployed people (%)), V5 (particulate matter including $PM_{2.5}$ and $PM_{10}$), V6 (nitrogen dioxide ($NO_2$)), V8 (sulfur dioxide ($SO_2$)), V9 (neighborhood deprivation index (%)), and V10 (road intersection density) consistently emerged as significant across model iterations, highlighting their influence as the most critical predictors in our next analysis and MLA predictions (Table 3).

**Table 3.** Stepwise selected model summary.

| Predictor | Estimate | Std. Error | z Value | Pr (>\|z\|) |
|-----------|----------|------------|---------|-------------|
| V1  | $1.97 \times 10^{-5}$ | $3.07 \times 10^{-6}$ | 6.433758  | $1.24 \times 10^{-10}$ *** |
| V4  | $-0.07422$            | 0.034637              | $-2.14278$ | 0.032131 * |
| V5  | 1.404977              | 0.470555              | 2.985788  | 0.002828 ** |
| V6  | 1865.001              | 413.0592              | 4.515094  | $6.33 \times 10^{-6}$ *** |
| V8  | 4250.563              | 1240.331              | 3.426957  | 0.00061 *** |
| V9  | $-0.00491$            | 0.002075              | $-2.36894$ | 0.017839 * |
| V10 | 0.000556              | 0.000138              | 4.041288  | $5.32 \times 10^{-5}$ *** |

Significance codes: 0—'***', 0.001—'**', 0.01—'*', 0.05—'.', 0.1—' ', 1, AIC = 1210.

### 3.4. Results and Performance of MLAs

The evaluation of various machine learning algorithms (MLAs) for predicting the response variable "asthma cases" across different neighborhoods offers insights into the performance of Random Forest (RF), Gradient Boosting Machine (GBM), and XGBoost. The evaluation metrics, including RMSE (root mean squared error), R-squared, MAE (mean absolute error), explained variance (EV), and Moran's I, help assess each model's fit to the training data and its ability to generalize to unseen test data, as well as the spatial randomness of residuals. These metrics are universally applicable and highly relevant for RF, GBM, and XGBoost regression models. In this study, we used 20% of the spatial data for testing. The MLA diagnostics are summarized in Table 4.

**Table 4.** Summary of MLA diagnostics.

| MLAs | RMSE | | R-Squared | | MAE | | EV | | Moran's I |
|---|---|---|---|---|---|---|---|---|---|
| | (Train) | (Test) | (Train) | (Test) | (Train) | (Test) | (Train) | (Test) | (Train) |
| RF | 0.56 | 1.08 | 0.96 | 0.75 | 0.40 | 0.84 | 1 | 0.74 | 0.29 |
| GBM | 0.56 | 1.07 | 0.95 | 0.76 | 0.43 | 0.88 | 0.95 | 0.75 | 0.17 |
| XGBoost | 0.22 | 1.21 | 0.99 | 0.69 | 0.16 | 0.91 | 0.99 | 0.68 | 0.12 |

The Random Forest algorithm demonstrates moderate performance, with a training RMSE of 0.56 and a test RMSE of 1.08, indicating a good fit on the training data but reduced performance on the test data. The R-squared values are 0.96 for the training set and 0.75 for the test set, suggesting that the algorithm explains a significant portion of the variance in the response variable, though less effectively on the test data. The MAE values are 0.40 for training and 0.84 for testing, highlighting higher prediction errors on unseen data. The explained variance (EV) values of 1.00 for training and 0.74 for testing further indicate that the algorithm performs well on the training data but may not generalize as effectively. The Moran's I value of 0.29 indicates moderate spatial autocorrelation in the residuals.

The Gradient Boosting algorithm performs slightly better, with a training RMSE of 0.56 and a test RMSE of 1.07, suggesting strong generalization compared to the Random Forest algorithm. The R-squared values are 0.95 for training and 0.76 for testing, reflecting good explanatory power for the variance in the data. The MAE values of 0.43 for training and 0.88 for testing indicate slightly lower prediction errors than the Random Forest algorithm. Additionally, the EV values of 0.95 for training and 0.75 for testing suggest effective pattern capture without substantial overfitting. The Moran's I value of 0.17 suggests lower spatial autocorrelation in the residuals compared to the Random Forest.

The XGBoost algorithm achieves the lowest training RMSE of 0.22 and the highest training R-squared of 0.99, indicating an almost perfect fit on the training data. However, the test RMSE of 1.21 and test R-squared of 0.69 suggest a notable decline in performance on the test data, implying potential overfitting. The MAE values are 0.16 for training and 0.91 for testing, showing minimal errors on the training set but higher errors on the test set. The EV values of 0.99 for training and 0.68 for testing further confirm that XGBoost may be overfitting to the training data. Additionally, Moran's I value of 0.12 indicates the lowest level of spatial autocorrelation in the residuals among the three algorithms, suggesting that XGBoost may handle spatial dependencies more effectively.

Among the Random Forest (RF), Gradient Boosting Machine (GBM), and XGBoost algorithms evaluated, GBM emerged as the best performer. It achieved the lowest test RMSE (1.07), high R-squared values (0.95 for training and 0.76 for testing), and strong explained variance (EV) values (0.95 for training and 0.75 for testing), indicating a superior balance between training and test performance. The algorithm also exhibited minimal

spatial autocorrelation in residuals, with a Moran's I value of 0.17, making it the most robust and suitable for predicting asthma cases in this dataset.

The XGBoost algorithm, while showing the lowest training RMSE (0.22) and highest training R-squared (0.99), demonstrated a significant decline in performance on the test data (test RMSE of 1.21 and test R-squared of 0.69), suggesting potential overfitting. Its Moran's I value of 0.12 indicates the lowest spatial autocorrelation in residuals, which is a positive aspect. Still, the overall model performance on test data was not as strong as GBM. The Random Forest algorithm had a test RMSE of 1.08, with R-squared values of 0.96 for training and 0.75 for testing, showing fair performance. However, it had a higher level of spatial autocorrelation in residuals (Moran's I value of 0.29) compared to GBM and XGBoost.

In sum, the GBM algorithm is the most suitable for this regression task. It offers a well-balanced and robust performance across various metrics, including minimal spatial autocorrelation, making it the best choice among this dataset's three algorithms for predicting asthma cases.

The variable importance diagnostics from the Gradient Boosting Machine (GBM) algorithm using R-Studio indicate that V1 (6.49%), V5 (4.53%), V9 (6.86%), and V10 (36.53%) are the most influential predictors for asthma occurrence, with V10 being the most significant. In contrast, V4 (1.2%), V6 (2.25%), and V8 (2.28%) contribute minimally to the model's predictive power. Mapping the most essential variables provides a detailed view of asthma occurrence by highlighting spatial patterns and trends. Figure 5 displays the spatial distribution of the values of the most important predictors in the study area at a high-resolution scale (100-cell size) based on the Gradient Boosting algorithm, our best-fitting model.

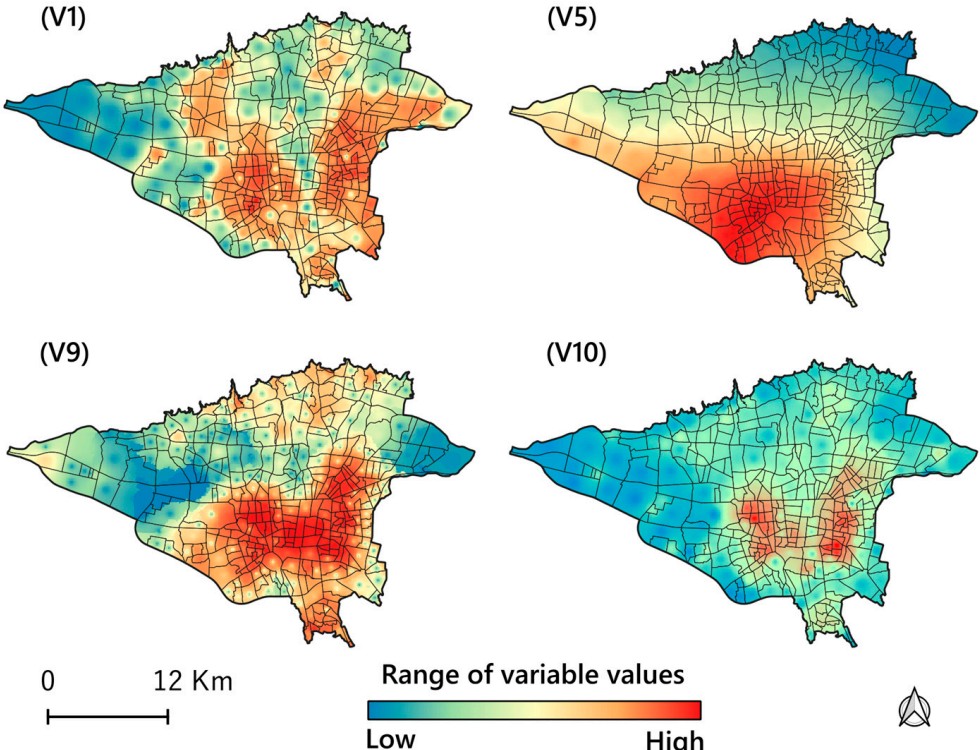

**Figure 5.** The Gradient Boosting algorithm determined spatial distribution maps of the most significant predictor values in the study area. Legend: V1: population density; V5: particulate matter including $PM_{2.5}$ and $PM_{10}$; V9: neighborhood deprivation index (%), which shows the most deprived areas; V10: road intersection density per square kilometers.

### 3.5. Visualizing Risk Prediction of Disease

The primary outcome of our study is a risk map developed using key predictors to assess the probability of asthma in the study area (Figure 6). Utilizing our best-fitting Gradient Boosting Machine (GBM) algorithm, validated with appropriate metrics, we generated a risk prediction map to evaluate the likelihood of asthma occurrence. To assess the model's accuracy, we employed a bivariate choropleth map in ArcGIS Pro version 3.2 [105]. This map compares the actual vs. predicted risk of asthma occurrence across various urban neighborhoods as determined by the GBM algorithm. The map demonstrates a strong agreement ($R^2 = 0.92$) between the model's predicted values and the observed asthma incident count per neighborhood. In a scatter plot, the $R^2$ value, or the coefficient of determination, measures how well the independent variable(s) explain the variance in the dependent variable. $R^2$ ranges from 0 to 1, where a value closer to 1 indicates that the data points fit the regression line well, meaning a strong relationship between the variables. Red line represents the identity line, indicating perfect prediction where observed values exactly match predicted values. Conversely, an $R^2$ closer to 0 suggests a weak relationship, with the data points more widely scattered around the red line.

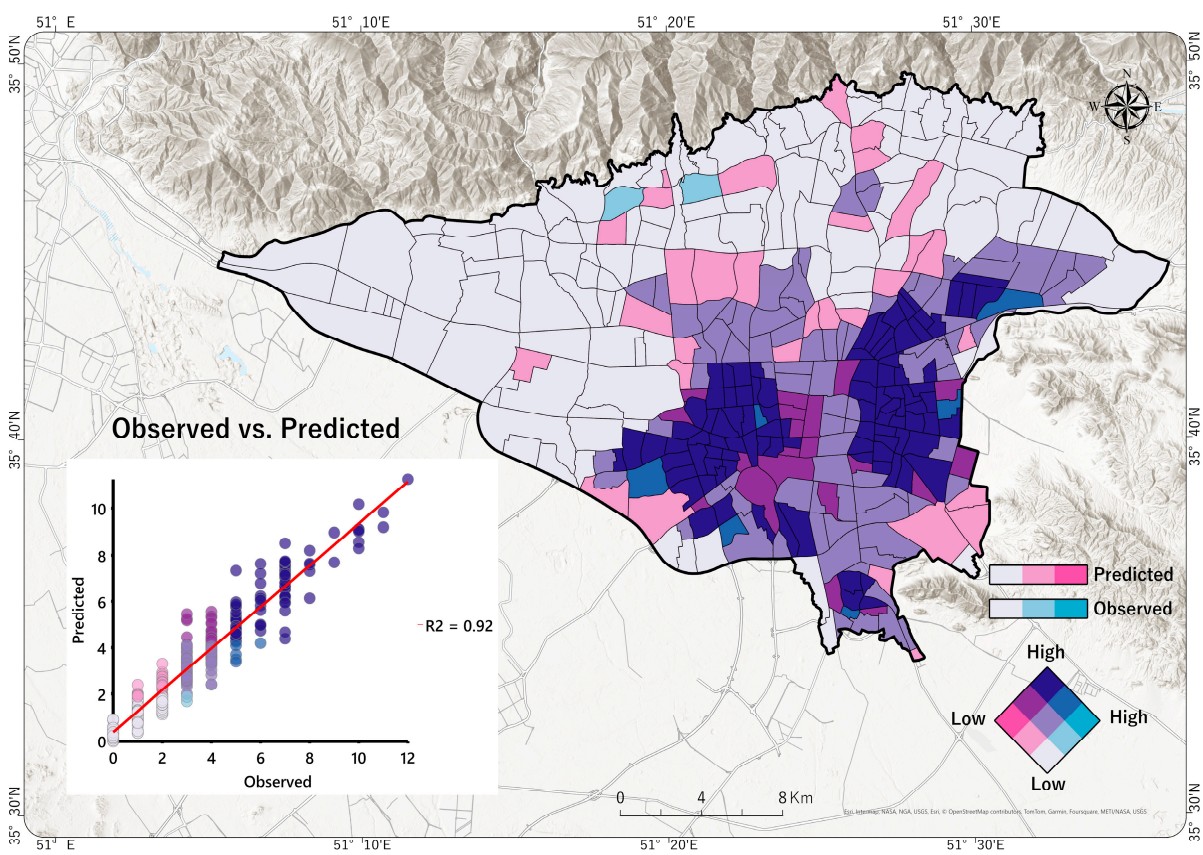

**Figure 6.** Spatial distribution map of locations with varying asthma occurrence risk probability levels within the study area.

Created in QGIS (QGIS Development Team, 2024), the map is designed for a straightforward interpretation, with high-risk neighborhoods highlighted. Based on the GBM model's results (Figure 6), around 164 neighborhoods (46.85%), home to an estimated 4,400,000 people (52% of the city's total population), are identified as high-risk areas, with predicted values exceeding the mean of approximately 3.42. The map indicates that the southwest and southeast regions near the city center have the highest risk of asthma occurrence. These insights can guide public health interventions and resource allocation, ensuring that high-risk areas receive the necessary attention to mitigate asthma risks.

## 4. Discussion

This study revealed significant age-related differences in asthma prevalence and outcomes, with a notable relationship between age groups and asthma incidence. Our findings showed that elderly patients with asthma are at a higher risk for morbidity and mortality from their asthma than younger patients, as corroborated by previous studies [106–108]. Asthma morbidity and mortality are higher in the elderly due to underdiagnosis, co-morbidities, and physiological changes like reduced lung elasticity and muscle strength. Immunosenescence, characterized by diminished immune responses and increased systemic inflammation that occurs with age, exacerbates asthma and infection risks [107,108]. Underutilization and reduced effectiveness of inhaled corticosteroids, coupled with airway remodeling, further elevate asthma risks. Additionally, age-related changes in lung structure, such as decreased chest wall compliance and increased airway obstruction, contribute to the severity of asthma in older adults [107,108].

In contrast, according to our findings, gender did not significantly correlate with asthma prevalence. Asthma prevalence is similar across genders due to balancing factors: boys have higher rates in childhood, while women have higher rates in adulthood due to hormonal influences and symptom reporting [109,110]. These opposing trends result in comparable overall prevalence. Genetic and environmental factors contributing to asthma risk also do not show strong gender bias, further equalizing prevalence rates between males and females [109,110].

Seasonal trends were observed, with most hospital admissions occurring from February to July, highlighting potential environmental or lifestyle factors influencing asthma exacerbations. According to previous studies [111–114], respiratory infections such as colds and flu are more common during these months, triggering asthma exacerbations [111]. Additionally, springtime increases pollen levels from trees, grasses, and flowers, leading to heightened allergic reactions in asthmatics. Changes in weather, including sudden temperature fluctuations and humidity [112,113] and increasing air pollutants [114], can also aggravate asthma symptoms. These combined factors contribute to the higher rate of asthma hospital admissions during this period.

Applying KDE analysis to 350 neighborhoods revealed significant spatial variability in asthma cases, with nearly half experiencing elevated asthma rates. The "hot spot" analysis further identified clusters of high and low asthma incidence, with notable concentrations of hot spots in the west and east of the city center. These results suggest localized factors affecting asthma prevalence and underscore the need to uncover underlying causes and guide targeted public health interventions. Asthma occurrences vary across large urban areas due to various environmental and socioeconomic factors and air pollutants, with some of these factors explored in this study. Higher asthma occurrences in specific neighborhoods are linked to elevated levels of air pollutants resulting from drought, traffic, and industrial activities [26]. Poor urban design, characterized by limited green spaces, combined with socioeconomic disparities, further impacts asthma occurrences, as lower-income areas face higher exposure to pollutants and have reduced access to healthcare [28,29]. Additionally, local climate and weather patterns can exacerbate asthma symptoms, leading to significant spatial differences in asthma occurrences within urban settings [112–114].

Using a validated and robust NBRM that exhibited no issues with overdispersion or multicollinearity, we identified population density (V1), proportion of unemployed people (V4), particulate matter (including $PM_{2.5}$ and $PM_{10}$) (V5), nitrogen dioxide ($NO_2$) (V6), sulfur dioxide ($SO_2$) (V8), neighborhood deprivation (V9), and road intersection density (V10) as the most significant predictors of asthma distribution (Table 3). These results underscore the critical influence of environmental characteristics (including built environment characteristics), air pollutants, and socioeconomic conditions on asthma

prevalence. We will discuss these influential factors in detail in the following sections, as they were identified as the most significant predictors by our robust machine learning algorithm analysis results.

Among the MLAs evaluated for predicting asthma cases, the GBM emerged as the top performer. It achieved the lowest prediction errors, highest R-squared values, and superior explained variance, indicating a strong balance between training and test performance. Additionally, the GBM algorithm demonstrated minimal spatial autocorrelation in residuals, making it the most robust and suitable algorithm for this dataset. The GBM findings highlighted road intersection density (V10), neighborhood deprivation index (V9), population density (V1), and particulate matter (V5) as the most influential predictors of asthma occurrence, with road intersection density being the most significant. In contrast, sulfur dioxide (V8), nitrogen dioxide (V6), and the proportion of unemployed people (V4) contributed minimally to the model's predictive power. This underscores the reliability and effectiveness of the GBM algorithm, making it a valuable tool for similar predictive tasks in the future.

According to the GBM algorithm, road intersection density (V10) predicts asthma cases significantly. Increased traffic congestion at intersections leads to higher emissions of pollutants such as particulate matter and nitrogen dioxide, exacerbating respiratory conditions and reducing air quality, contributing to the development and worsening of asthma symptoms [26]. Urban neighborhood deprivation (V9) is linked to asthma due to various interconnected factors, such as limited access to healthcare, higher pollution levels, poor housing conditions, increased stress, and greater exposure to environmental hazards. These conditions contribute to higher rates of respiratory problems, including asthma, in deprived and impoverished areas [21–23]. Population density (V1) is also associated with asthma due to higher pollution levels, increased exposure to allergens, and more frequent respiratory infections in densely populated areas. These factors collectively contribute to a higher prevalence of asthma [7,8]. Particulate matter ($PM_{2.5}$ and $PM_{10}$) (V5) is significantly associated with asthma. Fine particles ($PM_{2.5}$) can penetrate the lungs, causing inflammation and aggravating cardiovascular and respiratory conditions [14]. Increased levels of $PM_{2.5}$ are linked to a 2–3% rise in asthma symptoms among children. Similarly, elevated $PM_{10}$ concentrations are correlated with more frequent emergency room visits and hospital admissions for asthma, underscoring the substantial impact of particulate matter on asthma prevalence and severity [14,15,32–34].

The primary outcome of this study was the creation of a risk map using the Gradient Boosting Machine (GBM) algorithm to evaluate asthma occurrence across different neighborhoods. This map compared predicted asthma risks with observed cases and identified several high-risk areas, notably in the southwest and southeast zones near the city center. Compared to findings from previous studies [42–44], our use of various machine learning algorithms (MLAs) and a broader set of predictors enabled us to predict areas with the highest disease risk with greater reliability and accuracy. This approach confirmed earlier results and provided more nuanced insights into the spatial distribution of asthma risk, enhancing the robustness and precision of our predictive modeling.

### 4.1. Strengths, Limitations, and Future Directions

This study utilized GIS, remote sensing (RS), and ensemble machine learning algorithms, specifically GBM, to predict asthma-prone areas in urban settings. Identifying key predictors such as population density, particulate matter ($PM_{2.5}$ and $PM_{10}$), neighborhood deprivation index, and road intersection density yields a detailed risk map of Tehran's high-risk areas. These findings offer critical insights for targeted public health interventions, assisting community planners and administrators in managing asthma and

optimizing resource allocation. However, several limitations must be acknowledged. The study may not fully account for age-related complexities, gender differences, or all the environmental and lifestyle factors that influence asthma. Spatial variability suggests that localized factors influence prevalence, but the study may not account for all contributing variables. Reliance on specific data sources, such as satellite imagery and census data, may result in biases in spatial resolution, temporal variability, and data coverage. For example, remote sensing data may have limitations in accurately capturing fine-scale variations in urban environments, whereas census data may not fully reflect population dynamics or specific subpopulations. Furthermore, excluding factors such as urban heat islands and socioeconomic disparities risks overlooking essential determinants. These data-related biases may limit the results' generalizability, so caution is advised when applying these findings to other settings with different environmental or social characteristics.

Future research should incorporate more comprehensive data and additional variables to enhance predictive accuracy. Moreover, while machine learning algorithms are robust and predictive, they are computationally intensive and prone to overfitting, require extensive tuning, and can face obstacles of interpretability, outliers, and imbalanced data.

*4.2. Policy Implications*

The study's conclusions have ramifications for public health initiatives and urban development strategies meant to reduce the prevalence of asthma. The reported regional variations in Tehran's asthma prevalence highlight the need for a focused and localized strategy. Strict air quality monitoring, emission control in busy locations, and the encouragement of cleaner modes of transportation, especially in impoverished and highly populated neighborhoods, are all essential components of successful programs. Green space and buffer zone integration must be prioritized in urban development to lower pollution exposure and enhance air quality.

Addressing inequities in asthma outcomes requires improved access to healthcare services in high-risk communities and more public health surveillance. Additionally, the study highlights the need to address age-related vulnerabilities, particularly the increased risks older populations face due to comorbidities and decreased lung function. Tailored healthcare interventions, such as early diagnosis and better management strategies, are crucial to meeting their needs. Public awareness campaigns and infrastructure investments aimed at reducing traffic congestion can help minimize environmental triggers of asthma. Furthermore, addressing socioeconomic disparities through improved housing, education, and economic development initiatives can play a key role in mitigating the burden of asthma.

Important insights into the function of essential predictors such as particulate matter, neighborhood deprivation, and traffic intersection density were obtained by applying machine learning models, especially GBM. These results highlight the need for evidence-based decisions to efficiently distribute resources and create interventions focusing on high-risk locations. Metropolitan planners and legislators may promote healthier and more resilient areas by enacting egalitarian, data-driven, and locally relevant policies.

## 5. Conclusions

In this study, we integrated GIS, remote sensing (RS), and ensemble machine learning algorithms to predict asthma-prone areas in urban environments. Our results indicate that ensemble machine learning algorithms effectively identify asthma risk areas, with the Gradient Boosting Machine (GBM) algorithm demonstrating superior accuracy compared to other algorithms. Key predictors in our model were population density, particulate matter ($PM_{2.5}$ and $PM_{10}$), neighborhood deprivation index, and road intersection density.

The resulting asthma risk map highlighted higher-risk areas in the southern and western parts of Tehran near the city center, where increased population density and transportation contribute significantly to air pollution levels. Such risk maps provide valuable tools for community planners and administrators to manage and mitigate asthma in these regions. Additionally, our findings offer essential insights for guiding public health interventions and optimizing resource allocation to address asthma risks in the most affected neighborhoods.

**Supplementary Materials:** The following supporting information can be downloaded at: https://www.mdpi.com/article/10.3390/ijgi14030105/s1, Table S1: Data.

**Author Contributions:** Conceptualization, Alireza Mohammadi; methodology, Alireza Mohammadi; software, Alireza Mohammadi; validation, Alireza Mohammadi; formal analysis, Alireza Mohammadi; investigation, Elahe Pishgar; resources, Elahe Pishgar; data curation, Alireza Mohammadi; writing—original draft preparation, Alireza Mohammadi; writing—review and editing, Juan Aguilera; visualization, Alireza Mohammadi; supervision, Alireza Mohammadi; project administration, Alireza Mohammadi; funding acquisition, Alireza Mohammadi. All authors have read and agreed to the published version of the manuscript.

**Funding:** This research was funded by the University of Mohaghegh Ardabili, Iran, under grant number 14341. The funder provided financial support for the study, and the grant number has been verified for accuracy.

**Data Availability Statement:** The original contributions presented in this study are included in the Supplementary Material. Further inquiries can be directed to the corresponding author.

**Acknowledgments:** The authors thank the Masih Daneshvari Hospital (Tehran) for making the data accessible. The authors express gratitude to the University of Mohaghegh Ardabili (Iran), for extending the opportunity to conduct this research. Moreover, the authors wish to convey their profound gratitude to the editor-in-chief for their supportive comments and suggestions.

**Conflicts of Interest:** The authors declare no conflicts of interest.

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
