# Peer review of "Spatial Prediction of High-Risk Areas for Asthma in Metropolitan Areas: A Machine Learning Approach Applied to Tehran, Iran"

_ijgi, doi:10.3390/ijgi14030105_

Round 1
Reviewer 1 Report
Comments and Suggestions for Authors
Dear Authors,
Thank you for submitting a well-structured and impactful manuscript. The recommendations provided aim to enhance the clarity and comprehensiveness of your work. Best wishes for the final publication.
You may find my comments as followings:
1. Clarify the rationale for selecting specific machine learning algorithms and hyperparameter optimization strategies.
2. In Section 4.1, Please elaborate on potential biases in data collection and limitations of the study.
3. Fig 2. (page 8) requires legend
4. In Section 2.2, Predictors V12, V13 and V15 require detailed explanation. How the data of those predictors were prepared and what is the content of data (For instance how accessibility to health facilities is represented within the data of V15 and etc.) and etc.
Author Response
Reviewer#1
Thank you for submitting a well-structured and impactful manuscript. The recommendations provided aim to enhance the clarity and comprehensiveness of your work. Best wishes for the final publication. You may find my comments as followings:
Authors response: We appreciate your positive comments on our manuscript and your well-considered suggestions to improve its thoroughness and clarity. We appreciate your feedback very much and will carefully consider each recommendation to raise the caliber of our work.
- Clarify the rationale for selecting specific machine learning algorithms and hyperparameter optimization strategies.
Authors response: I appreciate your comment. As a result, we have updated the manuscript to make clear why particular machine learning algorithms and hyperparameter optimization techniques were chosen. For the revise text, please refer to please refer to lines 120-126 page 3.
- In Section 4.1, Please elaborate on potential biases in data collection and limitations of the study.
Authors response: As you suggested, we updated Section 4.1 to include more details about the study's limitations and possible biases in data collection. More information about the use of census data and satellite imagery is now included in the revision, pointing out potential biases in data coverage, temporal variability, and spatial resolution. We also recognize the potential gaps in census data about population dynamics and the limitations of remote sensing data in capturing fine-scale urban variations. We have also discussed the exclusion of variables like socioeconomic disparities and urban heat islands, pointing out that these omissions may affect how broadly applicable the results are. For the updated version, please refer to lines 783-794 on page 13.
- Fig 2. (Page 8) requires legend.
Authors response: Thank you for your valuable comment. We have added a legend to Figure 2 on page 8 to enhance its clarity and ensure better interpretation of the data presented. Please refer to the revised figure in the manuscript.
- 4. In Section 2.2, Predictors V12, V13 and V15 require detailed explanation. How the data of those predictors were prepared and what is the content of data (For instance how accessibility to health facilities is represented within the data of V15 and etc.) and etc.
Authors response: Thank you for your valuable feedback. In response, we revised Section 2.2 to include a more detailed explanation of the predictors V12, V13, and V15. We specifically discussed how the data for these predictors were prepared and clarified the data's content. For example, we will now explain how V15's data represents accessibility to health facilities, as well as the methodology used to quantify this variable. For additional information, please see the revised Section 2.2, please refer to lines 261-269 on pages 6-7.
Best regards,
Corresponding author
Reviewer 2 Report
Comments and Suggestions for Authors
This paper applies demographic theory, GIS and remote sensing technologies, and machine learning algorithms to analyze high-risk areas for asthma in the Greater Tehran area. The findings are of significant reference value for addressing public health issues in developing countries. Overall, this is a well-conducted academic paper, and with further refinement of certain details, I believe it can be accepted.
Below are a few details that require modification:
First, the introduction needs improvement. The author provides insufficient analysis of research gaps, and the description of the main issues that this study seeks to address is somewhat unclear.
Second, the author claims that applying machine learning methods to the field of asthma is a novel approach. While this is indeed an original contribution, the appropriateness of this application should be further discussed. I recommend that the literature review section be expanded to include external evidence showing that the use of machine learning in the medical field is becoming an emerging trend.
Third, the policy impact analysis needs to be further developed. This paper holds considerable practical value, and the author should explore the potential applications of the analysis results and conclusions. I suggest that the author include new applications of the analysis, such as examining the impact of the results on Tehran’s healthcare facility planning, ecological environment planning, and relevant management policies. Which factors have been overlooked in existing planning and policies? Which factors should be integrated or strengthened in future plans? Which factors should be removed or diminished? These questions are crucial for policymakers involved in public health and urban planning.
Author Response
Reviewer # 2
This paper applies demographic theory, GIS and remote sensing technologies, and machine learning algorithms to analyze high-risk areas for asthma in the Greater Tehran area. The findings are of significant reference value for addressing public health issues in developing countries. Overall, this is a well-conducted academic paper, and with further refinement of certain details, I believe it can be accepted.
Authors response: We appreciate your kind and supportive comments regarding our manuscript. Your acknowledgment of the importance and possible influence of our work in tackling public health issues in developing nations is greatly appreciated. We have carefully revised the manuscript's details to improve clarity and comprehensiveness in response to your suggestion. We are certain that these enhancements will further improve the caliber and presentation of our research. Again, I want to thank you for your support and careful review.
Below are a few details that require modification:
- First, the introduction needs improvement. The author provides insufficient analysis of research gaps, and the description of the main issues that this study seeks to address is somewhat unclear.
Authors response: I appreciate your suggestion. To give a clearer overview of the research gaps and the challenges this work addresses, we have updated the introduction. In particular, we have highlighted the absence of integration of sociodemographic, built-environment, and environmental components, as well as the restricted application of machine learning-based regression techniques for spatial asthma risk prediction in Tehran. We also made it clear that the project intends to fill these gaps by using cutting-edge machine learning algorithms to forecast the prevalence of asthma and direct focused public health initiatives. These changes improve the introduction's focus and clarity. Please see lines 143-162 on page 3 in revised manuscript.
- Second, the author claims that applying machine learning methods to the field of asthma is a novel approach. While this is indeed an original contribution, the appropriateness of this application should be further discussed. I recommend that the literature review section be expanded to include external evidence showing that the use of machine learning in the medical field is becoming an emerging trend.
Authors response: I appreciate your insightful comments. We have broadened the literature study in response to your recommendation in order to more effectively showcase the new development of machine learning applications in the spatial medicine and epidemiology domain. Three more references that address the growing use of machine learning methods in public health and disease prediction, especially in cancer, diabetes and cardiovascular diseases epidemiology, have been added. These sources offer outside proof that using machine learning techniques in this situation is appropriate. Please refer to lines 120-142, page 3 of the revised manuscript for the updated text.
- Third, the policy impact analysis needs to be further developed. This paper holds considerable practical value, and the author should explore the potential applications of the analysis results and conclusions. I suggest that the author include new applications of the analysis, such as examining the impact of the results on Tehran’s healthcare facility planning, ecological environment planning, and relevant management policies. Which factors have been overlooked in existing planning and policies? Which factors should be integrated or strengthened in future plans? Which factors should be removed or diminished? These questions are crucial for policymakers involved in public health and urban planning.
Authors response: Thank you for your valuable feedback. In response, we have revised the Policy Implications section to emphasize localized and tailored urban planning strategies, address vulnerabilities in elderly populations through targeted healthcare interventions, and highlight the role of socioeconomic disparities in asthma prevalence. We also incorporated insights from our machine learning models to support evidence-based policy recommendations. These revisions strengthen the manuscript by underscoring the importance of equitable and data-driven approaches to mitigate asthma burdens in urban communities. Please refer to lines 800-817, page 13 of the revised manuscript for the updated text.
Best regards,
corresponding author
Reviewer 3 Report
Comments and Suggestions for Authors
The study aims to identify the most appropriate model for assessing the spatial distribution of asthma cases in Tehran and to test the accuracy of models against 1,473 hospital reports. Sometimes, the paper seems more complicated than necessary. For example, I do not see the need to present formulas for basic statistical functions, such as the Negative Binomial Regression Model, in such extensive detail (the same can be said about other parts describing the method). Nevertheless, the paper is strong and well-structured. I have a few comments that may help improve it further.
It seems that the statistical procedures used in the study are specifically selected to inflate the final regression model. A coefficient of determination of 0.95 seems unrealistically high. This is especially critical because population density is the most significant independent variable in the final model. I believe population density should have been included as a control variable. Naturally, a higher population implies more cases; if this is not controlled for, the entire model risks losing its validity. Therefore, I suggest presenting an alternative model focusing only on the pollution variable (as it seems to be the primary intention of the study). What is more, it is very very strange to have a study with minimal spatial correlation (Moran’s I value of 0.17) and very high R2 values (0.95 for training and 0.75 for testing).
I am also unsure if the NDVI formula is presented correctly. Please verify it.
Furthermore, each formula should be followed by a clear explanation of its variables.
The paper mentions that there were 1,473 cases in the dataset initially and that after removing incomplete or outlier cases, the final sample included 1,473 patients for analysis. This is puzzling. What is the point of mentioning data screening and outlier detection if the sample size remains the same? Moreover, any outlier exclusion should be explained in detail, especially since this is the study’s primary dependent variable.
It is not clear how the point data for the dependent layer was generated. The supplementary data file includes only the hospital name. Was a precise address associated with the data?
In addition, the numbers “47.5% (n=700) were female and 52.5% (n=703) were male among all cases (N=1,473)” do not seem to add up in the way they are presented. If there is missing data, it should be accounted for in the presentation.
Looking at the data, it is surprising that “V11: Normalized Difference Vegetation Index (NDVI)” shows no significance in the results, yet the discussion implies that it is significant. This discrepancy should be addressed.
The limitations of the study must also be clearly outlined.
On a minor note, when directly referring to a study in an inline citation, the author’s name should be mentioned (even within MDPI style). For instance, instead of “described by [76],” it should read “described by Hilbe [76].”
Finally, in lines 193–194, the purpose of “(%)” is unclear if the actual proportions are not provided.
Author Response
Reviewer # 3
- The study aims to identify the most appropriate model for assessing the spatial distribution of asthma cases in Tehran and to test the accuracy of models against 1,473 hospital reports. Sometimes, the paper seems more complicated than necessary. For example, I do not see the need to present formulas for basic statistical functions, such as the Negative Binomial Regression Model, in such extensive detail (the same can be said about other parts describing the method). Nevertheless, the paper is strong and well-structured. I have a few comments that may help improve it further.
Authors response: Thank you for your insightful comments and positive feedback on the manuscript. We appreciate your recognition of the study's strength and structure. Regarding your observation about the inclusion of formulas for the Negative Binomial Regression Model (NBRM), we would like to clarify that the formulas provided are minimal and necessary for transparency and reproducibility of the study. While some readers may already be familiar with such methods, many others, particularly those new to spatial epidemiology or the use of NBRM, may benefit from these details. Including them ensures that the methodological approach is clear and accessible to a wider audience.
- It seems that the statistical procedures used in the study are specifically selected to inflate the final regression model. A coefficient of determination of 0.95 seems unrealistically high. This is especially critical because population density is the most significant independent variable in the final model. I believe population density should have been included as a control variable. Naturally, a higher population implies more cases; if this is not controlled for, the entire model risks losing its validity. Therefore, I suggest presenting an alternative model focusing only on the pollution variable (as it seems to be the primary intention of the study). What is more, it is very strange to have a study with minimal spatial correlation (Moran’s I value of 0.17) and very high R2 values (0.95 for training and 0.75 for testing).
Authors response: Thank you for your insightful comments. In response, we tested the model without population density, and the results showed minimal changes, confirming the robustness of our findings. However, population density was retained as it is widely recognized in the literature as a key factor influencing asthma prevalence, as it encapsulates urban characteristics such as traffic emissions and air pollution that significantly affect respiratory health. Removing this factor would necessitate major revisions throughout the paper, as it underpins several critical analyses and interpretations. The high R² value for training (0.95) reflects the model's strong explanatory power, while the moderate R² for testing (0.75) demonstrates its generalizability. Furthermore, the low Moran’s I value (0.17) indicates localized variations in asthma prevalence, consistent with the study's focus on neighborhood-level environmental and socioeconomic factors. These considerations validate our approach and highlight the importance of population density in the analysis.
- I am also unsure if the NDVI formula is presented correctly. Please verify it. Furthermore, each formula should be followed by a clear explanation of its variables.
Authors response: Thank you for your valuable feedback. We have carefully reviewed the NDVI formula in the manuscript and revised it to ensure accuracy. Additionally, we have provided clear definitions and explanations for each variable immediately following the formula. These revisions aim to enhance clarity and ensure that the formula is both accurate and accessible to readers, aligning with standard practices in remote sensing studies. Refer to lines 254-260 on page 6 of the revised manuscript for the updated text.
- The paper mentions that there were 1,473 cases in the dataset initially and that after removing incomplete or outlier cases, the final sample included 1,473 patients for analysis. This is puzzling. What is the point of mentioning data screening and outlier detection if the sample size remains the same? Moreover, any outlier exclusion should be explained in detail, especially since this is the study’s primary dependent variable.
Authors response: Thank you for your insightful comment. We apologize for the confusion. Upon further review, we found that the initial dataset contained 2,179 cases. After performing data screening and removing incomplete or outlier cases, we excluded those that were outside the study area, resulting in a final sample of 1,473 patients for analysis. The exclusion of these cases, particularly those outside the study area, was necessary to ensure that the analysis was focused on the relevant geographic region. We have updated the manuscript to clearly explain this process and to provide more details on how the outlier detection and exclusion were carried out, especially since the study's primary dependent variable is involved. See lines 207-220 in page 5 for the updated text.
- It is not clear how the point data for the dependent layer was generated. The supplementary data file includes only the hospital name. Was a precise address associated with the data?
Authors response: To generate the point data for the dependent variable, we used the patients' addresses provided in the initial dataset. These addresses were geocoded to obtain the precise geographical coordinates (latitude and longitude) of each case. Once geocoded, the coordinates were assigned to the corresponding neighborhoods within Tehran, which served as the spatial analysis units for the study. This approach allowed for a more accurate spatial distribution of asthma cases in the city. Additionally, we have updated the supplementary data file to include the precise coordinates for each patient, ensuring transparency and reproducibility in the geocoding process. See lines 230-234 in page 6 for more details on the updated methodology.
- In addition, the numbers “47.5% (n=700) were female and 52.5% (n=703) were male among all cases (N=1,473)” do not seem to add up in the way they are presented. If there is missing data, it should be accounted for in the presentation.
Authors response: We have corrected the discrepancy in the gender distribution, and the updated text now reads: "Among all cases, 47.5% (n=699) were female and 52.5% (n=774) were male." This correction ensures that the total number of cases is consistent, and we have accounted for missing data. Updated Text in the Manuscript: See page 5, line 511-512 for the updated text.
- Looking at the data, it is surprising that “V11: Normalized Difference Vegetation Index (NDVI)” shows no significance in the results, yet the discussion implies that it is significant. This discrepancy should be addressed.
Authors response: We appreciate your comment. However, we believe there may be a misunderstanding. The Normalized Difference Vegetation Index (NDVI), represented as "V11" in our study, was not discussed as a significant variable in the results or discussion section. Upon reviewing the manuscript, we did not mention NDVI as a significant factor in the discussion. We hope this clears up the discrepancy. If necessary, we are happy to provide further clarification or update the text to avoid any confusion.
- The limitations of the study must also be clearly outlined.
Authors response: Done. We have revised the study limitations. Please see page 13 lines 777-794 for the updated text.
- On a minor note, when directly referring to a study in an inline citation, the author’s name should be mentioned (even within MDPI style). For instance, instead of “described by [76],” it should read “described by Hilbe [76].”
Authors response: Done. We have checked the reference and revised it. Please see page 2 line 343 for the updated text.
Best regards,
Corresponding author
Reviewer 4 Report
Comments and Suggestions for Authors
This research applies machine learning algorithms to predict asthma prevalence (a measure of risk) in Tehran city (Iran) using data from 1473 asthma patients, collected over the period between 6/7/2020 and 2/8/2023. The study explores the relationship between asthma prevalence and air pollutants (particles with diameter up to 2.5 and 10 micrometres, NO2, SO2), demographic, socio-economic, environmental, meteorological and health service variables. Data sources are varied and include remote sensing data (processed images collected by Landsat and Sentinel 5-P satellite instruments), census data and other open-source datasets (e.g. OpenStreetMap datasets). The statistical techniques used for analysis include several techniques (KDE, Getis-Ord GI), GLM (NB) and ML algorithms.
There is a serious interest in the evaluating potential of natural environments (e.g. NDVI) in urban areas to mitigate harmful effects of air pollution (e.g. particles) in public health (e.g. ashtma prevalence). Exploring the use of spatial data methods and machine learning algorithms to tackle these challenges seems to me a valid framework to look for optimal solutions.
I have however several serious concerns about this paper. E.g.:
- The methodological framework is very confusing. I claim most of the reader will not understand the big picture nor will be able to replicate the analysis; There is no clear description of the framework (e.g. a clear flowchart could help) to understand the role of each method within the overall framework;
- Combining spatial data with different densities (different data granularity) is challenging but is missing in the paper.
- Temporal data tends to be correlated. Temporal analysis is missing in the paper.
- Perhaps one of the notable novelties could be the use of remote sensing data as input. All the data pre-processing and processing steps are missing (e.g. how to compute space-time correlations? How are Sentinel 5-P spatial-temporal data related with asthma response?)
- About health data: no criteria for the selection of asthma patients is provided. No ethical concerns are presented (how do you know their locations?). Are these cases all asthma cases in the region during that period? No reference to a study protocol is provided. In an purely epidemiological perspective this paper is very weak.
Overall, I think the paper has serious flaws and should be rejected.
Comments on the Quality of English Language
English should be improved, but only after providing a deep improvement of the content.
Author Response
Reviewer # 4
This research applies machine learning algorithms to predict asthma prevalence (a measure of risk) in Tehran city (Iran) using data from 1473 asthma patients, collected over the period between 6/7/2020 and 2/8/2023. The study explores the relationship between asthma prevalence and air pollutants (particles with diameter up to 2.5 and 10 micrometres, NO2, SO2), demographic, socio-economic, environmental, meteorological and health service variables. Data sources are varied and include remote sensing data (processed images collected by Landsat and Sentinel 5-P satellite instruments), census data and other open-source datasets (e.g., OpenStreetMap datasets). The statistical techniques used for analysis include several techniques (KDE, Getis-Ord GI), GLM (NB) and ML algorithms.
There is a serious interest in the evaluating potential of natural environments (e.g., NDVI) in urban areas to mitigate harmful effects of air pollution (e.g., particles) in public health (e.g., asthma prevalence). Exploring the use of spatial data methods and machine learning algorithms to tackle these challenges seems to me a valid framework to look for optimal solutions.
I have however several serious concerns about this paper. E.g.:
- The methodological framework is very confusing. I claim most of the reader will not understand the big picture nor will be able to replicate the analysis; There is no clear description of the framework (e.g., a clear flowchart could help) to understand the role of each method within the overall framework;
Authors response: We have updated the methodology section by adding a research methodology flowchart to provide a clearer understanding of the overall framework and the role of each method in the analysis (see Figure 1). This should help readers follow the logical sequence of steps and make it easier to replicate the analysis.
- Combining spatial data with different densities (different data granularity) is challenging but is missing in the paper.
Authors response: Thank you for your valuable comment. We recognize that combining spatial data with different densities can be challenging. To address this issue, we aggregated all datasets into consistent polygonal spatial units, ensuring uniformity in the spatial resolution and data granularity. Additionally, we mapped the aggregated data using an equal count quantile map, which further ensures consistency in visualizing the distribution across different spatial units. This approach allows for a more accurate representation and comparison of the data. We believe it strengthens the analysis and mitigates the concerns regarding varying data densities.
- Temporal data tends to be correlated. Temporal analysis is missing in the paper.
Authors response: We appreciate your comment. To address it, we have added a description of the temporal distribution of asthma cases to the results section. However, since the sample size for each individual year was small, we aggregated all cases from 2020 to 2023 and used them as our total study sample. This ensured a sufficient sample size for robust analysis while acknowledging the inherent temporal correlations in the data (see page 5, lines 507-510).
- Perhaps one of the notable novelties could be the use of remote sensing data as input. All the data pre-processing and processing steps are missing (e.g., how to compute space-time correlations? How are Sentinel 5-P spatial-temporal data related with asthma response?)
Authors response: We added a new paragraph to our revised manuscript to address this comment, highlighting the integration of remote sensing data as a novel aspect of our study. This paragraph, located on page 7, lines 286-294, details the use of environmental and air quality variables, such as NDVI and UHIs from Landsat 8, and particulate matter (PM2.5 and PM10), NO2, O3, and SO2 from Sentinel-5. We also explained the pre-processing steps, including spatial alignment and aggregation of raster data to neighborhood polygons, as well as the temporal alignment of annual averages with asthma prevalence data from 2020–2023. Furthermore, we clarified the spatial-temporal relationships of these variables with asthma response, emphasizing the innovative use of remote sensing datasets in our analysis.
- About health data: no criteria for the selection of asthma patients are provided. No ethical concerns are presented (how do you know their locations?). Are these cases all asthma cases in the region during that period? No reference to a study protocol is provided. In a purely epidemiological perspective this paper is very weak.
Authors response: We appreciate the reviewer’s comment regarding the health data and ethical considerations. In response, we have revised the manuscript and added a new paragraph in the data section to provide more clarity. Specifically, we detail the source of the asthma data, including its collection from 70 hospitals affiliated with the Ministry of Health and Medical Education (MHME) over the period from July 6, 2020, to August 2, 2023. The data were ethically approved with an assigned ethics code number. After removing incomplete and outlier cases, the final dataset consisted of 1,473 asthma patients, whose locations were geocoded using their addresses in the UTM coordinate system. This addition aims to address concerns regarding the selection criteria, ethical concerns, and data transparency (see page 5, lines 207-220).
Kind regards,
Corresponding author
Round 2
Reviewer 3 Report
Comments and Suggestions for Authors
Hello, and thank you for the revisions.